# Hyperprogressive Disease: Main Features and Key Controversies

**DOI:** 10.3390/ijms22073736

**Published:** 2021-04-03

**Authors:** Hugo Arasanz, Miren Zuazo, Ana Bocanegra, Luisa Chocarro, Ester Blanco, Maite Martínez, Idoia Morilla, Gonzalo Fernández, Lucía Teijeira, Pilar Morente, Miriam Echaide, Natalia Castro, Leticia Fernández, Maider Garnica, Pablo Ramos, David Escors, Grazyna Kochan, Ruth Vera

**Affiliations:** 1Oncoimmunology Group, Navarrabiomed, Fundación Miguel Servet-Public University of Navarre, IdiSNA, 31008 Pamplona, Spain; mzuazo@alumni.unav.es (M.Z.); ana.bocanegra.gondan@navarra.es (A.B.); luisa.chocarro.deerauso@navarra.es (L.C.); ester.blanco.palmeiro@navarra.es (E.B.); gfhinojal@gmail.com (G.F.); pilar.morente.sancho@navarra.es (P.M.); miriam.echaide.gorriz@navarra.es (M.E.); leticia.fernandez.rubio@navarra.es (L.F.); maider.garnica.suberviola@navarra.es (M.G.); pablo.ramos.castellanos@navarra.es (P.R.); grazyna.kochan@navarra.es (G.K.); 2Department of Medical Oncology, Complejo Hospitalario de Navarra-IdiSNA, 31008 Pamplona, Spain; maite.martinez.aguillo@navarra.es (M.M.); idoia.morilla.ruiz@navarra.es (I.M.); lucia.teijeira.sanchez@navarra.es (L.T.); natalia.castro.unanua@navarra.es (N.C.)

**Keywords:** hyperprogressive disease, immunotherapy, checkpoint inhibitors, cancer, solid tumors

## Abstract

Along with the positioning of immunotherapy as a preferential treatment for a wide variety of neoplasms, a new pattern of response consisting in a sudden acceleration of tumor growth has been described. This phenomenon has received the name of “hyperprogressive disease”, and several definitions have been proposed for its identification, most of them relying on radiological criteria. However, due to the fact that the cellular and molecular mechanisms have not been elucidated yet, there is still some debate regarding whether this fast progression is induced by immunotherapy or only reflects the natural course of some highly aggressive neoplasms. Moreover, contradictory results of trials including patients with different cancer types suggest that both the incidence, the associated factors and the implications regarding prognosis might differ depending on tumor histology. This article intends to review the main publications regarding this matter and critically approach the most controversial aspects.

## 1. Introduction

Hyperprogressive disease (HPD) is an adverse outcome of immunotherapy consisting in an acceleration of tumor growth, often accompanied by prompt clinical deterioration. The fundamental difference between HPD and immune-related adverse events (irAEs), which also might cause severe symptoms and a decline in the patient’s general condition, is that the latter induce damage through an autoimmune response, usually tissue- or organ-specific. By contrast, HPD is a caused by a direct stimulus for tumor growth and the deterioration is due to unspecific organ damage caused by cancer progression.

The first report of this phenomenon took place in 2016 in an NSCLC patient receiving nivolumab [1]. Since then, the number of scientific publications and reports has multiplied, reflecting the increasing interest in the medical community. Whether HPD is a specific phenomenon caused by immunotherapy or only a manifestation of the natural history of some tumors is still a matter of debate. Indeed, it is not only present in patients undergoing immunotherapy but also in chemotherapy treatments. Nevertheless, two retrospective studies found a superior incidence in immunotherapy cohorts compared with chemotherapy cohorts [2,3]. This study strongly suggests that hyperprogression is treatment-dependent, and that it may have a strong underlying immune component.

Unfortunately, a definitive answer to this question might be difficult to reach at this moment. Oncology, and especially immuno-oncology, is, at the present moment, a very dynamic field, and on many occasions research into the biological mechanisms is outrun by results from clinical trials. This situation is highly challenging for translational research as the availability of homogenous cohorts of patients, which would facilitate the study of a relatively uncommon outcome such as HPD, is obviously limited by new advances in cancer care. On the other hand, some discoveries might become quickly outdated as soon as the standard of care is renewed.

In addition to this, there is no agreement on the methods used for the identification of HPD yet. Clinical criteria are straightforward, although poorly specific. On the other hand, radiological criteria are precise and quantifiable. However, two CT scans before the start of immunotherapy are required, which is an important limitation considering that immunotherapy is steadily advancing to frontline treatment. A consensus should be reached to make different studies comparable and confirm that their findings refer to equivalent clinical situations.

## 2. Definition

At the present moment, the mechanisms causing HPD are yet to be elucidated, and therefore the identification of this phenomenon is based on radiological criteria, often complemented with clinical parameters. Several criteria have been proposed in different studies, although the vast majority compare tumor growth during immunotherapy with the immediately previous period. The main divergences between studies rely on quantification of tumor volume and the threshold of growth rate selected for the diagnosis of HPD.

The first published work in the subject by Sâada-Bouzid et al. used the “tumor growth kinetics” variable (TGK) to identify hyperprogressors. TGK is calculated by taking the difference of the sum of the diameters of the target lesions, according to RECIST criteria, per time unit between two radiological tests. The authors considered HPD when the ratio between TGK during immunotherapy and pretreatment TGK (TGK_R_) was higher than 2 [4]. This study only included patients with head and neck squamous cell cancer (HNSCC). Hence, it should be taken into consideration that HNSCC patients tend to have non-measurable lesions and frequent local ganglionic dissemination, which will likely introduce a degree of uncertainty.

The model utilized in this study implies a linear tumor growth model, which is not an accurate representation of the actual tumor behavior in many cancer types. To address this problem, Champiat S et al. evaluated HPD in a cohort of 218 patients with different tumor types under immunotherapy with the so-called tumor growth rate (TGR), which assumes an exponential growth mechanism [5,6]. In this study, HPD was diagnosed when the TGR during treatment had increased at least 2-fold from the reference period. A slightly different criterion was used by Ferrara R et al., and HPD was defined as a percentage increase in TGR higher than 50% [2]. This criterion identified HPD patients more accurately in a retrospective cohort of 406 NSCLC, as demonstrated by finding greater differences in overall survival (OS) between progressors classified as HPD and non-HPD patients [7].

Other groups have combined clinical and radiological variables for the identification of HPD. However, these approaches rely on the resources of the medical center and introduce certain elements that might be subjected to the characteristics of the patient, thus limiting the objectivity of the criteria. In the studies by Kato S et al., HPD was defined by simultaneously considering time to treatment failure (TTF) < 2 months, > 50% increase in tumor burden and > 2-fold increase in tumor growth rate [8]. Patients in the study by Lo Russo G required three out of five criteria for being diagnosed with HPD, including TTF < 2 months, increase ≥ 50% in the sum of the diameter of target lesions, appearance of at least two new lesions in an affected organ, dissemination to a new organ or clinical deterioration to performance status (PS) ≥ 2 [9].

Finally, Matos I et al. eliminated the requirement of a pre-basal CT scan and proposed an HPD definition requiring progressive disease (PD) in the first 8 weeks with an increase ≥ 40% in the sum of the diameters of the target lesions, or new lesions in at least two different organs [10].

As the mechanisms underlying HPD are still poorly defined, the best criteria for its identification are still unclear. However, a recent study comparing the definitions by Kas B, Matos I, Champiat S and Sâada-Bouzid E pointed out a superiority of the latter two. A substantial agreement was detected between them, with a Cohen’s kappa index of 0.61 and a Jaccard similarity matrix of 55%. This study also highlighted the relevance of including a pre-baseline CT scan for a more accurate identification of HPD [11].

To sum up, even though there are minor discrepancies, most of the authors consider HPD as a swift acceleration of tumor growth and the threshold might be highly dependent on the tumor type. Criteria requiring a pre-baseline CT are probably more precise, as they are the only means to differentiate a genuine HPD from a fast-growing tumor. On the other hand, the assumption of exponential growth by TGR seems more representative of tumor behavior. Lastly, the use of complementary clinical criteria might be useful to identify some HPD patients who do not fulfill radiological criteria due to non-measurable lesions or infrequent patterns of progression, although at the risk of losing objectivity.

## 3. Epidemiology

The incidence of HPD is highly variable among the different studies, which might be conditioned by the evaluated tumor types and the criteria used for diagnosis. The neoplasms with higher incidence of HPD seem to be gastric adenocarcinoma, NSCLC and HNSCC, although the data are still scarce and there is substantial variability between studies [12]. A rate of 9% was observed for NSCLC patients under immunotherapy by Champiat S et al. [5], 13.8% by Ferrara R et al. [2] and 17.9% by Arasanz H et al. [13]. According to the treatment received by the patient cohorts, most of the studies focused on PD-1/PD-L1 blockade with monoclonal antibodies, although in many of them, some patients received these drugs in combination. In fact, 33% of HPD patients had received combination immunotherapy in the trial by Kanjanapan Y et al. [14], while the study by Matos I et al. found a higher incidence of HPD in patients receiving combination immunotherapy compared with monotherapy (15.4% vs. 4.4%, *p* = 0.004) [10].

Several clinical and analytical variables have been correlated with a higher risk for HPD, even though some of the studies present contradictory results.

The intrinsic characteristics of the patients have also been considered for HPD risk, which include advanced age [5,15], female sex [14] or performance status of 1–2 [16], which are associated with HPD. Contributions by characteristics related to the extent of the neoplasm have also been shown to be associated, such as liver metastases [16,17], more than two metastatic locations at diagnosis [2,17], high tumor burden [16] and locoregional relapse in HNSCC [4].

According to analytical alterations and circulating cells, the associations of neutrophils, neutrophil-to-lymphocyte ratio, C-reactive protein and lactate dehydrogenase (LDH) over the median with HPD have been proposed as well [17]. The study by Sasaki A et al. in gastric cancer patients treated with nivolumab found an increase in blood C-reactive protein and in absolute neutrophil numbers only in HPD [16]. A meta-analysis of nine studies confirmed an increased risk of HPD if serum LDH levels were above the upper normal limit (OR = 1.89, *p* = 0.043), and when the liver (OR = 3.33, *p* < 0.001) or more than two organs were affected (OR = 1.89, p<0.001). PD-L1 tumor expression was inversely correlated with HPD (OR=0.60, *p* = 0.044) [12].

Lastly, a prospective translational study quantified specific immune cell subpopulations from freshly retrieved peripheral blood by flow cytometry and correlated these cell populations with HPD. Interestingly, lower percentages of CD4^+^ CD27^−^ CD28^−^ cells before starting immunotherapies were associated with HPD [13]. Undergoing a similar approach, preliminary results from a translational study by Ferrara R et al. presented at the AACR Congress 2020 found higher levels of circulating immature low-density CD10^−^ CD66b^+^ neutrophils (LDNs) in HPD patients compared to non-HPD progressors (36.6% vs. 10.1%, *p* = 0.04) [18].

Genomic alterations found in the tumors but also harbored by the patient have also been linked to HPD. Kato S et al. found an association of *MDM2/MDM4* amplification and *EGFR* alteration with time to failure (TTF) < 2 months by multivariate analysis, in a retrospective cohort of 155 patients with different tumor types. All six patients with *MDM2/4* amplification, 87.5% (seven out of eight) with *EGFR* alterations and 75% (three out of four) with *DNMT3A* alterations had TTF < 2 months. Of those, 67%, 29% and 0%, respectively, fulfilled the criteria stipulated for HPD, consisting in TTF < 2 months, 50% increase in tumor burden and > 2-fold increase in progression pace [8].

A communication presented at the ESMO 2017 Congress also reported amplifications of *MDM2/MDM4*, *EGFR* and different genes located on chromosome 11q13. These were the somatic alterations most frequently found by NGS in four HPD patients with different tumors. The retrospective evaluation of 10 patients with these alterations treated with immunotherapy uncovered an HPD incidence of 66% for *MDM2/MDM4*, 50% for *EGFR* and 43% for 11q13 [19]. *MDM2* amplification was also found in a gastric cancer patient with HPD, while the non-HPD patients from this study did not have alterations in this gene [20]. A small prospective work with NSCLC patients receiving a combination of camrelizumab (anti-PD-1) and apatinib (antiangiogenic tyrosine-kinase inhibitor, TKI) found short progression free survival (PFS) in two patients with *EGFR* and *FGF4* amplification (1.9 and 2.2 months, respectively [21].

Finally, a recent publication reported an association between polymorphisms in the host genome and HPD, as defined by Sâada-Bouzid E et al. [4], in patients receiving anti-PD1/PD-L1 monotherapy. The authors of the study evaluated single-nucleotide polymorphisms (SNPs) in the genes *PD-1*, *PD-L1*, *IDO1* and *VEGFR2*. By multivariate analysis, OR was 15.36 for rs2282055 in *VEGFR2* and 17.73 for rs2282055 in *PD-L1* [15].

It should be noted that in most of the clinical trials, the study groups have been HPD and non-HPD patients, this latter group also including patients with response or stabilization to immunotherapy [5,15,16]. As some of these variables have been previously associated with poor response to immune-checkpoint inhibitors (ICI), these findings might be merely the reflection of this fact. It would be of relevance to confirm if these differences are also found when comparing HPD patients with non-HPD progressors, as explored in other works [2,10].

## 4. Biomarkers

HPD is considered to be a systemic phenomenon, so it is reasonable to expect that it could have a reflection in systemic biomarkers that may be detectable in peripheral blood. Two studies characterized the dynamics of circulating immune cells in peripheral blood and their correlation with HPD. Both of the studies included NSCLC patients receiving PD-1/PD-L1 blockade therapies.

Kim CG et al. found an association between elevated percentages of severely exhausted (TIGIT+) tumor-reactive CD8+ and HPD. A profile with decreased effector/memory (CCR7− CD45RA−) CD8+ T cells and increased TIGIT+ CD8+ T cells was restricted to HPD patients and not to other progressors. However, the low abundance of CCR7− CD8+ T cells could reflect the enrichment of CD45RA+ CCR7+ CD8+ T cells in HPD patients [17]. On the other hand, Arasanz H et al. found that an increase ≥ 30% in highly differentiated CD28neg CD27neg CD4 T cells (CD4 T_HD_ burst) between the first and the second cycle of immunotherapy identified HPD patients with a sensitivity of 70% and a specificity of 82% (AUC = 0.792). Patients with CD4 T_HD_ burst presented lower PFS than the rest of the patient cohort (6.29 vs. 9.86 weeks, *p* < 0.001) and a trend towards lower PFS when compared with non-HPD progressors [13].

Other studies have focused on alternative biomarkers such as tumor circulating DNA. A study with 56 patients monitored chromosomal instability (CI) from plasma cfDNA at different time points. A correlation of radiological response with a decrease in cfDNA before the second cycle was reported. This cohort included six cases of HPD and one patient with pseudoprogression. The changes in CI allowed an early and accurate identification of HPD in five out of six patients, even before routine radiological tests [22].

Radiomics has been recently explored as a novel approach to predict response to immunotherapy. This approach correlates imaging data with CD8 abundance in tumor samples by RNAseq [23,24,25,26,27]. Using this technique, four radiomics-based models were constructed in a study of digestive tumors in patients receiving anti-PD-1/PD-L1 immunotherapy. The model constructed from baseline and post-treatment CT characteristics had an AUC = 0.806, with a sensitivity of 83.3% and a specificity of 88.9% for the identification of responders. Among progressors, 3 features out of 220 were selected from a model relying on the maximum gray value from the baseline CT and first radiological evaluation, rendering an AUC of 0.877 for HPD detection [28]. More recently, a study in NSCLC patients treated with anti-PD-1/PD-L1 monotherapy identified a peritumoral texture feature and two vessel-related tortuosity features among 198 textural elements from tumor nodules and peripheral tissue to distinguish HPD with an impressive AUC value of 0.96 in the validation cohort. HPD patients had elevated values of the selected features compared with responders and non-HPD progressors, with a sensitivity of 100% and 81% specificity. Additionally, predicted HPD patients presented lower OS compared with non-HPD (20 vs. 38 months, *p* = 0.009), and the differences remained significant when compared with non-responders (HR 5.93, *p* < 0.0001) [29].

## 5. Mechanisms

Even though many clinical, analytical and genomic aspects have been associated with HPD, the discovery of cellular and molecular mechanisms explaining this unwanted phenomenon would strongly support its recognition as an existing adverse event. Thus far, two studies have uncovered some advances on this research subject. These studies describe alterations involving different cell types, suggesting that HPD is not a phenomenon with a single cause but a consequence of several events.

The first attempt to identify underlying immunological mechanisms of HPD was published in early 2019 and targeted myeloid populations. Lo Russo G et al. found a significant infiltration by M2-like CD163^+^ macrophages in all HPD patients in pretreatment tumor samples of NSCLC patients receiving immunotherapy. Using a murine model, the authors suggested that HPD was induced by anti-PD1 antibodies. This induction was found to be dependent on the Fc fragment, probably through its interaction with the Fcγ receptor of M2-like macrophages [9].

On the other hand, Kamada T et al. showed an enrichment in proliferating regulatory T cells (Tregs) in HPD patients, while this population was reduced in non-HPD. This study was carried out in tumor biopsies performed before and after treatment in gastric cancer patients. In agreement with high PD-1 expression in Tregs, the authors confirmed in vitro and in vivo that the immunosuppressive activity and proliferation of these cells were increased with anti-PD-1 antibodies. The authors proposed that CTLA-4-, OX-40- or CCR4-targeted therapies could be strategies to prevent HPD by depleting Tregs [20].

These two works are of great value, as for the first time, plausible mechanisms for HPD with in vitro and in vivo demonstrations have been proposed. However, both of them are based on the retrospective study of tumor biopsies and are limited to only one cancer type. The differences between the human and mouse immune systems, and particularly the lack of concordance of the FcRs, might question the findings of the study by Lo Russo G et al. [9]. In the work by Kamada T et al., the low number of HPD patients is a main concern [20]. A prospective validation in cancer patients would be required before confirming the two HPD mechanisms.

As mentioned before, different gene mutations have been associated with HPD. However, their mechanistic implication in HPD development is only hypothetical at most. It has been demonstrated that the EGFR pathway suppresses immune responses by activating Tregs, and these Tregs could then be further stimulated by PD-1 blockade [30]. In a study using a lung cancer xenograft mouse model, PD-1 blockade stimulated tumor proliferation and reduced apoptosis [31]. In addition, a mechanism inducing *JAK-STAT* activation via IFN-γ has been suggested. This pathway can further potentiate IRF-8 expression which might stimulate MDM2 expression [8]. However, a direct link from these immunosuppressive mechanisms to HPD has not yet been proven.

Whether these potential mechanisms leading to HPD, if finally confirmed, are tumor-dependent or not is still unknown, Figure 1. However, when clinical contexts where the patients receiving ICI are at higher risk of developing HPD are identified, combination approaches based on this information could provide a significant advance to prevent fast progression or death caused by HPD. 

## 6. Implications

At the present moment, HPD is only considered a radiological event induced by one or several mechanisms. As its identification is subject to the many limitations of radiological evaluation, the study of the implications regarding patients’ prognosis is of significant medical importance.

As expected, most of the clinical trials have shown lower progression-free survival (PFS) in HPD patients. In HNSCC patients, it was 2.5 months, and in non-HPD patients, (including 12% of responders) it was 3.4 months (*p* = 0.003). When applying irRECIST criteria, it was 2.9 vs. 5.1 months, respectively (*p* = 0.02) [4].

The study by Kanjanapan Y et al. including various tumor types and with 20% of patients receiving anti-PD-1/PD-L1 combined with other immune-stimulatory molecules found a significantly lower PFS for HPD patients (1.6 vs. 2.8 months, *p* < 0.001). Again, patients with partial response (8%) or disease stabilization (47%) were included in the non-HPD group. Interestingly, the proportion of patients receiving combination immunotherapy in the HPD group (33%) was comparable to the complete cohort, suggesting that HPD is not limited to anti-PD-1 monotherapy [14].

Two studies detected not only worse PFS but also OS in HPD patients. Kim CG et al. compared HPD with non-HPD progressors in NSCLC patients, finding lower PFS (19 vs. 48 days, HR 4.619; *p* < 0.001) and OS (50 vs. 205 days, HR 5.079; *p* < 0.001) [17]. This association with lower median PFS and median OS was also found by Sasaki A et al. (0.7 vs. 2.4 months, *p* < 0.001 and 2.3 months vs. NR, *p* < 0.001, respectively) and Aoki M (0.9 vs. 1.7 months, *p* = 0.004 and 2.1 vs. 5.5 months, *p* = 0.002) [3,16]. However, responding patients were again included in the non-HPD group. Moreover, no differences in PFS or OS were found by Aoki M et al. when comparing HPD with non-HPD progressors (0.9 vs. 0.8 months, *p* = 0.756 and 2.1 vs. 3.1 months, *p* = 0.168, respectively) [3].

Intriguingly, some trials have reported lower OS in HPD patients without inferior PFS. Two French studies found lower OS in HPD patients but without differences in PFS. The study by Champiat S et al. in a total number of 131 patients with NSCLC, renal cell carcinoma, HNSCC, urothelial cancer and Hodgkin lymphoma found a median OS of 4.6 months in HPD, while non-HPD patients had an OS of 7.6 months. However, this difference was not statistically significant (*p* = 0.19) [5]. In agreement with these results, but with a more homogeneous population, Ferrara R et al. found lower OS mostly in pretreated NSCLC patients with HPD within the first 6 weeks compared with non-HPD progressors (3.4 vs. 6.2 months, *p* = 0.003) [2]. The study by Matos I et al. showed lower OS for HPD patients compared with the remaining progressors (5.23 vs. 7.33 months, *p* = 0.004). However, it has to be noted that these results were obtained only after application of their HPD definition, which does not require the evaluation of TGR in the pretreatment period [10] [Table 1].

## 7. Future Strategies and Discussion

Vast evidence indicates that immunotherapy might have a deleterious effect on some cancer patients, even accelerating the spread of their disease. Further research is warranted to confirm the underlying mechanisms described thus far, and to explore other possible molecular processes inducing this paradoxical response to ICI. However, clinical trials do not usually include HPD based on radiological criteria as a pattern of response, thus making the evaluation of this phenomenon in large cohorts of patients difficult.

At this moment, the most effective and viable strategy for avoiding HPD is patient selection according to the forementioned characteristics associated with this outcome. Moreover, patients presenting some of these features, such as liver metastases, high serum LDH or high NLR, yield less benefit from immunotherapy [32,33]. This could be a feasible option when therapeutic alternatives with comparable efficacy are available, but otherwise it would be problematic to discard overall survival-prolonging drugs based on small retrospective trials.

As certain immune subpopulations have been identified as potentially responsible for HPD, its targeting with drug combinations could be a practical approach. However, it should be kept in mind that it is still not clear if the HPD mechanisms described do occur simultaneously or, on the contrary, are mutually exclusive. Furthermore, they could even be tissue-dependent, hence making this strategy much more complex.

The role of Tregs described by Kamada T et al. positions this population as a compelling target, but it is well known that its depletion induces fatal autoimmune disorders [20,34]. Interestingly, daclizumab, an anti-CD25 monoclonal antibody, was able to decrease and reprogram Tregs without unleashing significant autoimmunity in breast cancer patients, although lowering effector T (Teff) function due to IL-2 signaling blockade on these cells [35]. Moreover, the group of Quezada SA recently developed a CD25-blocking monoclonal antibody with an engineered Fc fragment (anti-CD25^NIB^), showing Treg depletion while conserving IL-2/STAT5 signaling on Teff [36]. Other molecules overexpressed by Tregs such as CCR4, CCR8, CTLA-4, OX40 and 4-1BB could also be proposed as targets [37,38]. However, in the case of CTLA-4 blockade, the studies by Kanjanapan Y et al. and Matos I et al. and a case report suggest that an anti-PD1 and anti-CTLA4 combination does not preclude HPD [10,14,39].

Finally, regarding HPD induced by the interplay with the Fcγ receptor of M2 macrophages, both engineering the Fc fragment of anti-PD1/PD-L1 antibodies or the blockade of M2 macrophages through deletion or reprogramming could be plausible strategies. Different compounds have been shown to reduce M2 TAMs levels and are currently under clinical evaluation. The disruption of the CCL2/CCR2 and CXCL12/CXCR4 interactions with antibodies limits monocyte recruitment from the blood and their accumulation. TAMs can also be depleted by the blockade of CSF-1R with different compounds such as PLX3397 or emactuzumab [40,41], as well as with chemotherapeutic drugs already used in other clinical contexts such as trabectedin [42].

Anyhow, before any experimental strategy to prevent HPD is proposed, additional research that identifies the cellular and molecular mechanisms underlying this phenomenon, as well as the clinical and pathological context where it occurs, is clearly required. A greater implication from immuno-oncology studies of promoters and awareness of this outcome from clinical researchers will be key to fulfilling this objective.

## Figures and Tables

**Figure 1 ijms-22-03736-f001:**
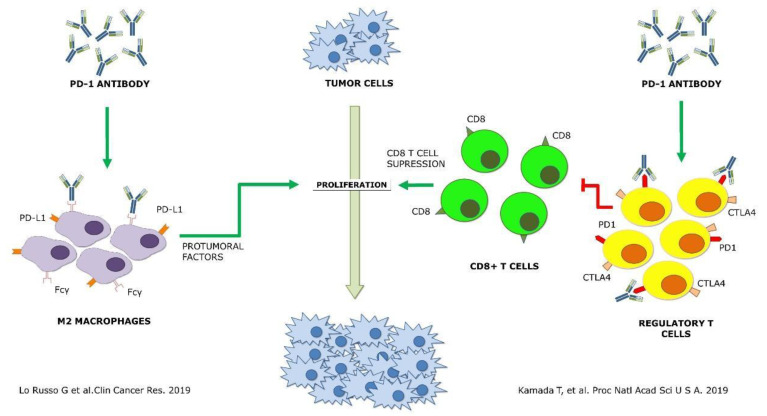
Schematic representation of the two proposed mechanisms of hyperprogressive disease up to date [9,20] Green arrows, agonist interaction. Red arrows, antagonist interaction.

**Table 1 ijms-22-03736-t001:** Summary of the most relevant works studying hyperprogressive disease. HNSCC: head and neck squamous cell cancer; LDH: lactate dehydrogenase; LDNs: low-density neutrophils; NLR: neutrophils-to-lymphocyte ratio; NSCLC: non-small cell lung cancer; PCR: protein C-reactive; PD: progressive disease; PS: performance status; SNP: single-nucleotide polymorphism; TGK: tumor growth kinetics; TGK_R_: tumor growth kinetics ratio; TGR: tumor growth ratio; TL: T lymphocyte; TTF: time to failure.

Authors	Tumor Type	Patients	HPD Definition	Incidence (%)	Biomarker
Arasanz H (2020) [13]	NSCLC	70	TGR ≥ 2	17.9%	Low CD4+ CD27− CD28− cells (blood) Increase CD4+ CD27− CD28− ≥ 30%
Champiat S (2017) [5]	Mixed Cohort	131	TGR ≥ 2	9%	Age ≥ 65
Ferrara R (2018) [2]	NSCLC	406	∆TGR > 50%	13.8%	> 2 metastatic sites
Ferrara R (2020) [18]	NSCLC	46	TGR ≥ 2 ∆TGR > 50%	9%	CD10^−^ CD66b^+^ LDNs (blood)
Kanjanapan Y (2019) [14]	Mixed Cohort	182	TGR ≥ 2	7%	Female sex
Kamada T (2019) [20]	Gastric Cancer	36	Combined: - TTF < 2 months - > 50% increase in tumor burden - 2x increase in TGR	11.1%	None
Kato S (2017) [8]	Mixed Cohort	155	Combined: - TTF < 2 months - > 50% increase in tumor burden - 2x increase in TGR	3.9%	MDM2 amplification EGFR alteration
Kim CG (2019) [17]	NSCLC	263	TGK_R_ ≥ 2 TGR ≥ 2 TTF < 2 mo	20.9% (TGK_R_) 20.5% (TGR) 37.3% (TTF)	≥2 metastatic locations Liver metastases Neutrophils (blood) NLR PCR (blood) LDH (blood) High CD8+ PD-1+ TIGIT + TL Low CD8+ CCR7− CD45RA− TL
Kim JY (2019) [12]	Mixed Cohort	1519	N/A	14.3%	LDH (blood) Liver metastases > 2 metastatic locations Low tumor PD-L1
Lo Russo G (2019) [9]	NSCLC	152	3 out of 5 of: - TTF < 2 months - ≥ 50% diameter increase - ≥ 2 new lesions - Spread to new organ - Decline to PS ≥ 2	25.7%	Tumor-infiltrating macrophages
Matos I (2020) [10]	Mixed Cohort	270	PD in the first 8 weeks and 1 of: - ≥ 40% diameter increase - new lesions in ≥ 2 new organs	10.7%	Liver metastases >2 metastatic locations
Refae S (2020) [15]	Mixed Cohort	98	TGK_R_ ≥ 2	14%	Age ≥ 70 *VEGFR2* SNP *PD-L1* SNP
Sâada-Bouzid E (2017) [4]	HNSCC	34	TGK_R_ ≥ 2	29%	Local relapse
Sasaki A (2019) [16]	Gastric Cancer	62	TGR ≥ 2	21%	PS 1-2 Liver metastases Large tumor lesions Neutrophils (blood) PCR (blood)
Singavi A (2017) [19]	Mixed Cohort	Not specified	Increase tumor size > 50% TGR ≥ 2	Not specified	*MDM2/4* amplification *EGFR* alterations
Vaidya P (2020) [29]	NSCLC	109	TGK_R_ ≥ 2	17.4%	Radiomic model (vessel tortuosity and peritumoral textures)
Weiss GJ (2017) [22]	Mixed Cohort	56	Not specified	10.7%	Chromosomal instability changes

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
