# Peer review of "Hyperprogressive Disease: Main Features and Key Controversies"

_ijms, 2021, doi:10.3390/ijms22073736_

Round 1

Reviewer 1 Report

The Authors of this review aimed to present current concepts on hyperprogressive process in malignancy. This article brings explanation of the subject as a whole. There is very informative Table, bibliography is current and numerous. The articles was divided into chapters, it facilitates understanding the problem. The main uncertainty is definition of the disease. And it will be good to present conclusion in the chapter “Definition”- a clear message to readers will be valuable. It can be a personal view of the Authors.

The Authors cite many studies, the bibliography is appropriately reach. However the value of many studies could be questionable. Maybe the Authors could add their own opinion? It concerns a chapter Mechanisms. This chapter is rather short and could be enriched. On the figure the mechanisms are simplified. The readers of this review can get an impression that immune reaction in cancer is one-dimensional. The population of macrophages in cancer milieu is very heterogenous, Tregs also are very complex cells. Thus, it will be worthy to underline that the scheme is only approximate. Do I well understand that Figure is a copy, not original Author’s figure? Recently, many papers try to explain HPD, my suggestion is to add new concepts and include in the figure, but it is only suggestion.

What about differential diagnosis of HPD with IrAE? Especially in the lung. Please add some sentences for clarification.

Minor comment- please explain all abbreviations in the description of the table.

Author Response

We really appreciate the kind commentaries and suggestions by Reviewer 1, as we think they improve the review and make it more attractive and easy-reading for other colleagues.

  1. The Authors of this review aimed to present current concepts on hyperprogressive process in malignancy. This article brings explanation of the subject as a whole. There is very informative Table, bibliography is current and numerous. The articles was divided into chapters, it facilitates understanding the problem. The main uncertainty is definition of the disease. And it will be good to present conclusion in the chapter “Definition”- a clear message to readers will be valuable. It can be a personal view of the Authors.

A paragraph including a brief conclusion and our personal view has been added.

  1. The Authors cite many studies, the bibliography is appropriately reach. However the value of many studies could be questionable. Maybe the Authors could add their own opinion? It concerns a chapter Mechanisms. This chapter is rather short and could be enriched.

We included the main limitations of the two works by Lo Russo G et al and Kamada T et al, and added our point of view.

  1. On the figure the mechanisms are simplified. The readers of this review can get an impression that immune reaction in cancer is one-dimensional. The population of macrophages in cancer milieu is very heterogenous, Tregs also are very complex cells. Thus, it will be worthy to underline that the scheme is only approximate. Do I well understand that Figure is a copy, not original Author’s figure? Recently, many papers try to explain HPD, my suggestion is to add new concepts and include in the figure, but it is only suggestion.

We agree with the reviewer, and we are thankful for this commentary. The figure is original and was designed for this review. Our intention with the figure was to present the hypothesis suggested by the only two studies focused on the mechanistics of HPD, and not to show a detailed overview of the immune response induced by immunotherapy, which is much more complex. We remarked that the representations is merely an scheme of the findings of two papers.

  1. What about differential diagnosis of HPD with IrAE? Especially in the lung. Please add some sentences for clarification.

A brief clarification has been included in the introduction.

  1. Minor comment- please explain all abbreviations in the description of the table:

An explanation of each abbreviation has been included.

Reviewer 2 Report

Very good review and focused on important problems asscociated with the use of immune checkpoints inhibitors.

Author Response

We really appreciate the reviewer's comments and support.